# A Similarity-Agnostic Reinforcement Learning Approach for Lead Optimization

## Abstract

Lead optimization in drug discovery is a pivotal phase in identifying promising drug candidates for further development. Traditionally, lead optimization in the machine learning community has been treated as a constraint optimization problem where methods like generative models and reinforcement learning (RL) have been widely employed. However, these methods often rely on molecular similarity metrics to define constraints, which poses significant challenges due to the inherently ambiguous nature of molecular similarity. In this work, we present a similarity-agnostic approach to lead optimization, which we term "Lead Optimization using Goal-conditioned Reinforcement Learning" or LOGRL. Contrary to conventional methods, LOGRL is uniquely trained on a distinct task: source-to-target path prediction. This allows LOGRL to produce molecules with significantly higher Tanimoto similarity to target molecules, even without direct exposure to this metric during training. Furthermore, we incorporate a beam search strategy during the molecule generation process. This strategy empowers us to generate a substantial number of candidate molecules, facilitating further curation to meet desired properties. Notably, our unique approach permits us to leverage the Euclidean distance between learned action representations as a surrogate for molecular similarity during beam search.

## 1 Introduction

Lead optimization is the final step of the drug discovery process where the goal is to maintain favourable properties of the lead candidate while improving on the deficiencies Hughes et al. (2011). In this work, we target the first stage of lead optimization which involves generating a number of molecules as alternatives to the lead candidate with similar structure but better properties. We henceforth refer to it as 'lead optimization' for brevity.

With recent advances in deep learning, a number of approaches have been introduced to solve the challenge of lead optimization. Recent approaches involve formulating lead opimization as a constraint optimization problem commonly using generative models (Yu et al., 2022; Bresson & Laurent, 2019; Kong et al., 2021; Ma & Zhang, 2021; Maziarka et al., 2020; Huang et al., 2022), etc. or reinforcement learning (You et al., 2018; Zhou et al., 2019; Işık & Tan, 2021; Ahn et al., 2020; Ståhl et al., 2019), etc.. In reinforcement learning (RL) methods, similarity is the metric often used for defining constraints. However, similarity in molecules is not a well-defined concept or, as Bender & Glen (2004) say, similarity *has a context*. Consequently, using any similarity metric as a constraint introduces bias into the system towards the intricacies of that similarity metric. Further, in RL methods for lead optimization, the similarity constraint is integrated as a part of the reward function. This introduces additional hyperparameters that are not only costly to tune, but also hard to optimize empirically (Hayes et al., 2022). Additionally, this induces several challenges well known in the multi-objective RL literature, such as Pareto dominance, conflicting objectives, questions on the linearity of reward design, etc. (Hayes et al., 2022). These problems become worse as the number of properties used in the reward function increases.

To overcome these challenges, we propose a similarity-agnostic method for lead optimization based on goal-conditioned RL that is trained on the related task of source-to-target path prediction. Further, we offload the task of property optimization from the training phase and suggest post-hoc curating the generated molecules. Towards this end, we also propose a generation strategy based on

beam-search that uses the Euclidean distance of the learned action representations for generating a diverse range of candidate molecules. An additional advantage of using this method is that it can be significantly cheaper (in time and compute resources) to generate a large number of molecules using the same model and then search for required properties than to train different models for each combination of the required properties.

To ensure the validity of molecules, we follow fragment replacement methods similar to Tan et al. (2022) but instead of replacing fragments on the lead candidate, we operate "reaction rules" on smaller starting molecules, mimicking a step-by-step drug creation process. These reaction rules, mined from USPTO-MIT (Jin et al., 2017) induce synthesisability into the design of the molecules and also provide a possible path of synthesis as part of the generation process.

We therefore separate the synthesisability, generation, and optimisation into different aspects of design and improve the individual capabilities of each part, thereby improving the whole process.

We summarise our contributions as follows:

1. We propose a novel strategy using goal-conditioned reinforcement learning to generate molecules similar to a given target molecule.

2. We propose a search strategy that separates the property optimization from the training and offloads it as a post-curation process, thereby simplifying the task of learning.

3. We mine "reaction rules" from USPTO-MIT (Jin et al., 2017) that increase the synthesisability of the proposed candidates as well as suggest a possible route of synthesis.

## 2 RELATED WORKS

We review related works that target lead optimization with an emphasis on reinforcement learning methods for a closer comparison.

You et al. (2018) use reinforcement learning to generate molecular graphs. To affix the distribution of their generated molecules similar to a set of target molecules, they utilize an adversarial reward along with the rewards for property optimization. However, this method requires a large number of target molecules to generate a reliable adversarial reward which is a hard expectation in real-world lead optimization scenarios. Zhou et al. (2019) train a DQN to generate molecule graphs and give rewards as a weighted combination of similarity and a property for optimisation. Işık & Tan (2021) build upon Zhou et al. (2019) and use graph networks for molecular representations while giving similar rewards as them. Ahn et al. (2020) also give similar rewards but they define their RL framework over the genetic algorithm operators mutation and crossover. Ståhl et al. (2019) utilize BiLSTM-based actor-critic architectures to learn which fragments to replace, and give rewards as a combination of indicator function that denotes property satisfaction and distribution of generated molecules that satisfy the property. They ensure the similarity of their generated molecules to the targets by replacing fragments with high similarity. Olivecrona et al. (2017) use RL to fine-tune an RNN by giving rewards for similarity to generate molecules similar to Celecoxib. Tan et al. (2022) train a transformer over SMILES and give rewards for a combination of SC score and a threshold-ed tanimoto similarity. All these methods use similarity-based rewards which introduces bias into their methodologies. Jeon & Kim (2020) also builds upon Zhou et al. (2019) but they give rewards for docking score along with property score. Though they do not use similarity-based rewards, their work still suffers from the challenges of multi-objective RL. Moreover, giving rewards for docking scores can be very expensive and infeasible for larger molecular spaces like ours. Another limitation of most of these methods is that synthesisability is not integrated into the generation process. It is sometimes induced by adding SA score (Ertl & Schuffenhauer, 2009) as another term in the reward function which only further exasperates the reward design problem.

Other works include sampling-based methods (Fu et al., 2021), search-based methods (Hartenfeller et al., 2012; Kawai et al., 2014; Sun et al., 2022), auto encoders and VAEs (Yu et al., 2022; Bresson & Laurent, 2019; Kong et al., 2021; Ma & Zhang, 2021), flow models (Luo et al., 2021; Kuznetsov & Polykovskiy, 2021; Shi et al., 2020; Zang & Wang, 2020; Ma & Zhang, 2021), GANs Maziarka et al. (2020), transformers (Piao et al., 2023), diffusion (Huang et al. (2023) and genetic algorithms (Lee et al., 2021).

## 3 BACKGROUND

### 3.1 REINFORCEMENT LEARNING

Reinforcement learning tasks are modelled as Markov Decision Processes (MDPs). An MDP is defined as a tuple of $\langle \mathcal{S}, \mathcal{A}, \mathcal{T}, \mathcal{R}, \gamma, \rho_0 \rangle$ where $\mathcal{S}$ is the set of states, $\mathcal{A}$ is the set of actions, $\mathcal{R} : \mathcal{S} \times \mathcal{A} \times \mathcal{S} \to \mathbb{R}$ is the reward function, $\mathcal{T} : \mathcal{S} \times \mathcal{A} \times \mathcal{S} \to [0, 1]$ is the transition probability distribution, $\gamma$ is the discount factor and $\rho_0 : \mathcal{S} \to [0, 1]$ is the probability distribution over initial states.

The learning objective of RL is to learn a policy $\pi : \mathcal{S} \times \mathcal{A} \to [0, 1]$ that maximizes the expected discounted return given by:

$$J(\pi) = \mathbb{E}_{\substack{a_t \sim \pi(\cdot|s_t), \\ s_{t+1} \sim \mathcal{T}(\cdot|s_t, a_t)}} \left[ \sum_t \gamma^t R(s_t, a_t, s_{t+1}) \right]$$

### 3.2 GOAL CONDITIONED REINFORCEMENT LEARNING

GCRL tasks are modelled as goal-augmented MDPs (GA-MDP). In GA-MDP, an additional tuple is defined $\langle \mathcal{G}, \rho_g \rangle$. $G$ denotes the set of goals and $\rho_g$ is the distribution of goals. The objective changes to learning a policy $\pi$ that maximizes the expected discounted return given by:

$$J(\pi) = \mathbb{E}_{\substack{a_t \sim \pi(\cdot|s_t), g \sim \rho_g, \\ s_{t+1} \sim \mathcal{T}(\cdot|s_t, a_t)}} \left[ \sum_t \gamma^t R(s_t, a_t, s_{t+1}, g) \right]$$

In our case, goals are from the same space as the set of states. So we have $\mathcal{G} \subseteq \mathcal{S}$.
There are two ways to give rewards in GCRL:

1. Binary rewards, in which a reward is given only if the algorithm has reached the goal state:

$$R = \begin{cases} 1, & \text{if } s_{t+1} = g \\ 0, & \text{otherwise.} \end{cases}$$

   This type of reward is the easiest to give without requiring to do any reward shaping but can lead to high sparsity in the reward space, resulting in challenges with learning (Tang & Kucukelbir, 2021).

2. Distance-based rewards, which reward the algorithm if it has gotten closer to the goal state:

$$R = -d(s_{t+1}, g),$$

   where $d$ is some distance measure between states and goals. This leads to dense rewards but changes the loss landscape of the objective leading to local optimas (Trott et al., 2019).

### 3.3 OFFLINE REINFORCEMENT LEARNING

In offline RL, the algorithm only has access to a static dataset $D_O = \langle s, a, s', r \rangle$ collected using some behaviour policy. In this setting, the algorithm is not allowed to interact with the environment to collect more samples.

We use offline RL to overcome the sparse reward problem in GCRL and improve sample efficiency. Hence, we do not address the challenges faced by offline RL in this work and leave that for future work.

## 4 METHODS

### 4.1 PROBLEM SETUP

We model the lead optimization task as source-to-target path prediction using the goal-conditioned reinforcement learning paradigm. States and goals are molecules, and actions are reaction rules in

the form of reaction signatures as described in section 4.2. A reaction rule applied on a reactant state results in a deterministic product. The agent is given rewards for reaching the goal. Thus, the GCRL agent is given different source-target molecule pairs and is tasked to learn to take actions that convert the source into the target molecule.

Due to the sparse reward problem when solving the task in an online GCRL setup, we opt to use an offline RL dataset with high reward trajectories to promote learning. Details on the offline data generation are provided in section 4.4.

## 4.2 REACTION RULES AS ACTIONS

For this study, 84,968 reaction rules were used that we mined from the USPTO-MIT dataset (Jin et al., 2017) according to Sankar et al. (2017). Each rule is a tuple of two subgraphs - the reactant's signature $r^{sig}$ and the product's signature $p^{sig}$. $r^{sig}$ and $p^{sig}$ correspond to the functional groups removed and added during a chemical reaction, along with neighbours up to 2 atoms away to denote the structures on which the rule is applicable. To determine the site of the reaction, the mining process also includes information about reaction centres, which are the atoms in the signatures where the subgraph addition or removal occurs. An example is shown in Figure 1.

Figure 1: **Reaction signatures**: The illustration depicts reaction signatures involved in the reaction transformation of $m_t$ to $m_{t+1}$. $r_t^{sig}$ is the reactant signature and $p_t^{sig}$ is the product signature. Centres are marked in yellow. The signatures are calculated as the changed subgraphs + neighbors up to 2 atoms away.

The applicability of an action is determined by the reactant's signature. Thereafter, given a reactant molecule, all those rules that have their reactant signature present in the molecule are considered to be applicable on it.

## 4.3 STATE AND ACTION REPRESENTATIONS

The state consists of two molecules: the source and the target. The action consists of two signatures: reactant signature and product signature. All of these are graph objects, so we use a Graph Isomorphism Network (GIN) (Xu et al., 2018) to encode each and create learnable representations.

## 4.4 OFFLINE RL DATASET

We generate an offline RL dataset using trajectories with high returns to overcome the sparse reward problem in GCRL. Details on the dataset generation process can be found in the supplementary material.

## 4.5 NEGATIVE ACTION SELECTION STRATEGY

The offline dataset only contains trajectories with high returns. Learning from only successful actions leads to policies that cannot discriminate between good and bad actions. Hence for each sample in the training batch, batch we uniformly randomly sample $l$ actions during runtime to constitute *negative samples*. These $l$ actions are selected from the list of all actions except the one in the high-return sample. Dynamically allowing the agent to see samples with low returns enhances the agent's distinguishing capabilities and improves performance.

---

**Algorithm 1** Pseudocode for training actor-critic

---

**Inputs**: offline dataset, action dataset, GIN, actor, critic, KNN
**Output**:
 1: **repeat**
 2:   Sample positive batch $B = \langle s, a, s', r \rangle$ from the offline dataset.
 3:   Use actor to predict $a'$ on $s$.
 4:   Collect negative batch $\bar{B} = \langle s, \bar{a}, s', \bar{r} \rangle$ using the negative action selection strategy from section 4.5.
 5:   Update critic with eq equation 2 using $B \cup \bar{B}$.
 6:   Update actor using PG loss from eq equation 1 using $B \cup \bar{B}$.
 7: **until** convergence

---

## 4.6 Rewards and Loss

Here we discuss the actor and critic's rewards, returns and loss functions.

As mentioned previously, training with an offline RL dataset addresses the sparse reward problem inherent in GCRL. This enabled us to use binary rewards in our design. However, according to our formulation, the PG loss is zero when the return is zero, hence we add an additional penalty to stimulate the actor's learning.
The reward function R is defined as:

$$R = \begin{cases} 1, & \text{if } s_{t+1} = g \\ -\frac{1}{2l}, & \text{if } s_{t+1} \neq g \text{ at the end of episode} \\ 0, & \text{otherwise.} \end{cases}$$

Due to our small episode length, we consider the undiscounted case. Using the reward function defined above, we can compute the return G as follows:

$$G = \begin{cases} 1, & \text{if } a_t \text{ leads to } g \\ -\frac{1}{2l}, & \text{otherwise.} \end{cases}$$

We use the standard policy gradient (PG) loss for actor $\pi_\theta$ and MSE for the critic $Q_\phi$:

$$\nabla_\theta \mathcal{L}(\theta) = \mathbb{E}_{\tau \sim \pi_\theta} \big[ G(\tau) \nabla_\theta log \pi_\theta(\tau) \big] \tag{1}$$

$$\mathcal{L}(\phi) = \frac{1}{2} \big( Q_\phi(\tau) - G(\tau) \big)^2 \tag{2}$$

Our choice of $R$ when $s_{t+1} \neq g$ at the end of episode is experimental. We found that under the condition $G(a_t \text{ leads to } g) \leq l * G(otherwise)$, the actor's performance starts to deteriorate. On investigation, we found that this is due to the cumulative magnitude of gradients from the negative return trajectories per positive trajectory being greater than that of the positive return trajectory. This would lead to a learning behaviour that directs the policy away from areas of negative returns even at the cost of moving away from areas with positive returns.

## 4.7 Training and Generation Algorithms

Algorithm 1 describes the pseudocode for the training algorithm. Figures 2(a), (b), (c) show pictorial descriptions of the training algorithm.

Algorithm 2 describes the pseudocode for generating molecules similar to the target molecule. Figure 2(d) shows the generation procedure.

Brief descriptions of the algorithms are provided in the supplementary material.

## 5 Experiments

In this section, we discuss our experimental setup and evaluation test-bed, including datasets, models, evaluation settings and metrics.

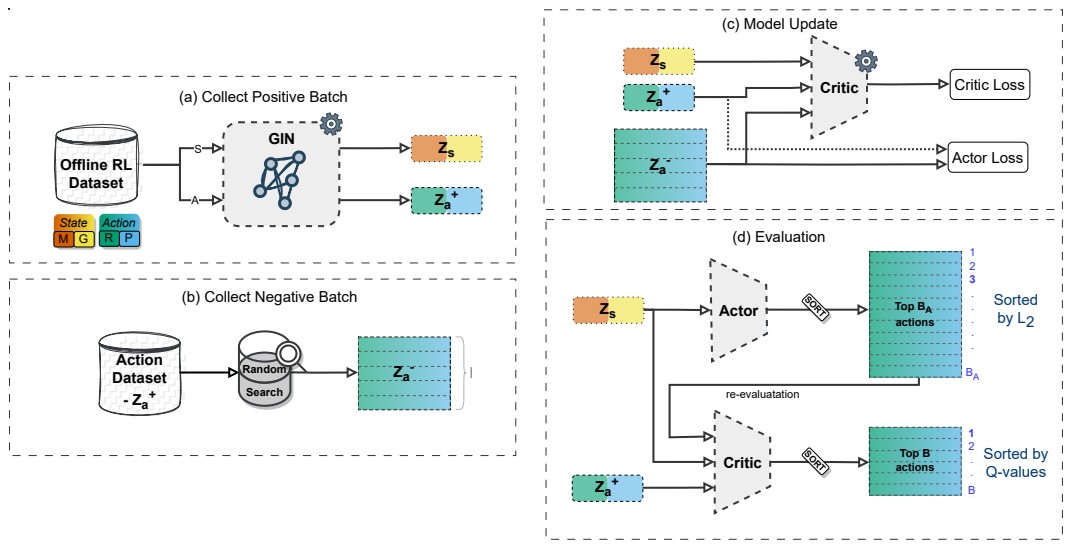

Figure 2: **Training and evaluation procedure for actor-critic:** (a) shows sampling of positive samples from the offline RL dataset. Each sample contains current molecule $m_t$, target molecule $m_n$, reactant signature $r_t^{sig}$ and product signature $p_t^{sig}$. The sample is passed through the embedding module to create $Z_s$ and $Z_a^+$, the state and positive action representations. (b) shows the creation of negative samples by randomly sampling action embeddings other than $Z_a^-$ (c) shows the inputs to the actor and critic losses - the actor loss is calculated over the action embeddings $Z_a^+$ and $Z_a^-$, and the critic loss is calculated over the Q value. The gradients are then used to update the two models. (d) shows the evaluation procedure. First, the actor's prediction is used to order the action dataset according to the Euclidean distance from the prediction to create to select the top $B_A$ actions. These are sent for re-evaluation to the critic. These are sorted by the critic's Q values and the top $B$ actions are returned.

---

**Algorithm 2** Pseudocode for generating molecules similar to target molecule

---

**Inputs**: list of source-target pairs $M_0$, steps $N$, GIN, actor, critic, action dataset, branching factor $B$, actor's branching factor $B_A$
**Output**:

 1: Get action embeddings $D_a$ using GIN.
 2: **for** $(i = 0; i < N; i + +)$ **do**
 3:     Init $M_{i+1}$ as an empty list
 4:     **for** each molecule pair $m$ in $M_i$ **do**
 5:         Predict action embeddings $Z$ on $m$ using actor.
 6:         Get top $B_A$ actions from action dataset by closest Euclidean distance between $Z$ and $D_a$.
 7:         Get Q values for $m$ and the $B_A$ actions using critic.
 8:         Get top $B$ actions from these $B_A$ according to the highest Q values
 9:         Get products by applying above $B$ actions on $m$'s source molecule and add it to $M_{i+1}$ along with $m$'s target.
10:     **end for**
11: **end for**
12: Return $M_N$

---

## 5.1 DATASETS

We generate an offline RL dataset with trajectories of length 5 as described in section 4.4 containing 100,000 samples, which we further subdivide into 80,000 and 20,000 for training and validation. The test dataset contains 20,000 samples for evaluation.

## 5.2 TRAINING SETUP

We use a Graph Isomorphism Network(GIN) (Xu et al., 2018) for molecular embeddings and an actor-critic model for learning the policy. We refer the reader to the supplementary document for details on the architecture of the models and hyperparameters used.

## 5.3 BASELINES

We compare our method with similarity-based optimization methods. Towards that effect, we train two baseline models in an online RL fashion using PPO (Schulman et al., 2017). The first baseline model is only given rewards for similarity, and the second is given rewards as a combination of similarity and QED. For computing molecular similarity, we use Tanimoto similarity between Morgan Fingerprints with radius 2 as Zhou et al. (2019). We refer to it simply as 'similarity' henceforth.

1. *S model* refers to the model trained to optimize similarity. Reward is given as:

$$r_t = sim(m_t, m_n)$$

   Where $m_t$ is the molecule at time $t$ and $m_n$ is the target molecule.

2. *Q+S model* refers to the model trained to optimize similarity and QED. We define the reward according to Zhou et al. (2019):

$$r_t = w \times sim(m_t, m_n) + (1 - \omega) \times QED(m_t)$$

   We choose $\omega = 0.4$ in our experiments as it is the highest $\omega$ in their work such that the relative improvement of QED is not centered close to zero.

Following Gottipati et al. (2020), we also train a Random Search baseline. In this baseline, initially, a random molecule is chosen $m_0$ from the set of start molecules and a random action is applied on it to produce $m_1$. This is repeated until a termination condition of the trajectory is met; either the maximum length of trajectory is achieved or no actions are applicable on the molecule.

Details on hyperparameters used for training are provided in the supplementary material.

## 5.4 EVALUATION

We evaluate each model on their generated molecules with targets as five trypsin inhibitors from Hartenfeller et al. (2012). Starting molecules were selected from Enamine Building Block catalogue [1] Global Stock. They were filtered using highest similarity between the starting molecules and targets, highest similarity between the scaffolds (using rdkit's *GetScaffoldForMol* function) of starting molecules and targets, and the maximum common subsequence (using rdkit's *FindMCS* function) between starting molecules and targets to select 200 unique starting molecules per target.

Then the generation procedure was run on all 1000 pairs as described in 4.7.

We compare the mean similarity and min, max and mean QED over 5 sets containing the top 1, 10, $10^2$, $10^3$ and $10^4$ of generated molecules by similarity. We also evaluate the set of molecules on validity, uniqueness and novelty from Brown et al. (2019).

Following Hartenfeller et al. (2012), we run the generation procedure for $N = 4$ timesteps and branching factor $B = 10$. We choose the actor's branching factor $B_A = 50$.

The Random Search baseline involves no training component and is directly used for the generation procedure. It is run 200 times for each target for a fair comparison with the other models.

## 6 RESULTS AND DISCUSSION

Table 1 shows the results for the three models: S model, Q+S model and LOGRL. Figure 3 shows the distributions of similarity and QED for all the models for the set of $10^4$ molecules with the highest similarity.

---

[1] https://enamine.net/building-blocks

| Method | Mols | Mean sim | QED Min | QED Mean | QED Max | Val | Uniq | Nov |
|---|---|---|---|---|---|---|---|---|
| **Random Search** | 1 | 0.477 | 0.483 | 0.483 | 0.483 | 1 | 1 | 1 |
| | 10 | 0.45 | 0.353 | 0.448 | 0.559 | 1 | 1 | 1 |
| | $10^2$ | 0.417 | 0.109 | 0.418 | 0.841 | 1 | 1 | 1 |
| | $10^3$ | 0.377 | 0.041 | 0.385 | 0.841 | 1 | 1 | 1 |
| | $10^4$ | 0.333 | 0.022 | 0.316 | 0.929 | 1 | 1 | 1 |
| **S model** | 1 | 0.694 | 0.289 | 0.289 | 0.289 | **1** | **1** | **1** |
| | 10 | 0.618 | 0.12 | 0.201 | 0.289 | **1** | **1** | **1** |
| | $10^2$ | 0.554 | 0.049 | 0.206 | **0.712** | **1** | 0.99 | **1** |
| | $10^3$ | 0.499 | 0.024 | 0.262 | **0.923** | **1** | 0.999 | **1** |
| | $10^4$ | 0.439 | 0.013 | **0.302** | **0.946** | **1** | 0.9988 | **1** |
| **Q+S model** | 1 | 0.703 | **0.294** | **0.294** | **0.294** | **1** | **1** | **1** |
| | 10 | 0.652 | **0.264** | **0.401** | **0.681** | **1** | **1** | **1** |
| | $10^2$ | 0.59 | **0.102** | **0.299** | 0.681 | **1** | **1** | **1** |
| | $10^3$ | 0.523 | 0.029 | 0.226 | 0.845 | **1** | 0.999 | **1** |
| | $10^4$ | 0.45 | 0.013 | 0.261 | 0.942 | **1** | 0.9997 | **1** |
| **LOGRL** | 1 | **0.791** | 0.262 | 0.262 | 0.262 | **1** | **1** | **1** |
| | 10 | **0.761** | 0.127 | 0.244 | 0.321 | **1** | **1** | **1** |
| | $10^2$ | **0.704** | 0.088 | 0.294 | 0.624 | **1** | **1** | **1** |
| | $10^3$ | **0.634** | **0.032** | **0.3** | 0.791 | **1** | **1** | **1** |
| | $10^4$ | **0.56** | **0.018** | 0.291 | 0.889 | **1** | **0.9999** | **1** |

Table 1: Table of results. The Three baseline models: Random Search, S model and Q+S model, and our proposed LOGRL model are evaluated for the generated molecules on the top $10^n$ molecules based on the highest similarity ('Mols' column). The 'Mean sim' column represents the mean similarity among the chosen molecules according to the 'Mols' column. 'QED Min', 'QED Mean', and 'QED Max' represent the min, mean, and max QED values from the same set of molecules. 'Val', 'Uniq', and 'Nov' represent validity, uniqueness, and novelty according to Brown et al. (2019) from these set of molecules. The metrics of Random Search are not highlighted as its purpose is to provide a base of comparison for other models and its generated molecules would not be lead optimization candidates.

**Similarity comparison: Random Search vs optimization methods -** The Random Search baseline results in molecules with low similarity values. This is natural since the baseline is not given any information about the target molecules. The significantly higher values of mean similarity of the other models indicate that the training of these models was successful and the generated molecules from these models can be further evaluated for lead optimization.

**Similarity comparison: LOGRL vs online RL baselines -** LOGRL outperforms both online RL baselines for mean similarity on all of the molecule sets. This shows that the similarity-agnostic model trained on the task of source-to-target path prediction better learns how to generate molecules similar to the target.

**Similarity comparison: *Q+S model* vs *S model* -** The two baseline models perform very similar to each other for mean similarity, with *Q+S model* slightly outperforming *S model* for all molecule sets. This was surprising as the *Q+S model* is given very less weight for similarity as compared to QED. But we found that Işık & Tan (2021)'s experiments sometimes show similar trends in their Table 3 where they show results for different values of $\omega$. We attribute it to some correlation between similarity and QED, particular to the source and target molecules.

For lead optimization, high similarity is a more restrictive constraint than a high QED. To test the utility of the generated molecules, we performed one more analysis. From the set of $10^4$ molecules, we tested the stats for all the molecules with QED $\geq 0.7$. LOGRL generated 73 such molecules with a mean similarity of 0.54 and a maximum similarity of 0.65. Q+S model, on the other hand, gener-

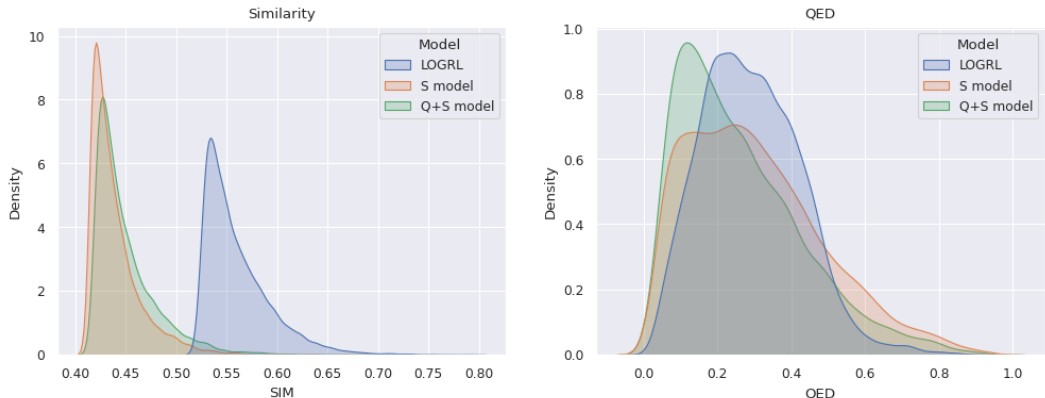

Figure 3: Distributions of Similarity and QED for the top 10000 molecules (by similarity) generated by the three models: S model, Q+S model and LOGRL.

ated 204 molecules with a mean similarity of 0.44 and max similarity of 0.53, which is lower than even the mean similarity of the molecules generated by LOGRL. This shows that though LOGRL generated less molecules with high QED, the molecules are more likely to be lead candidates than the Q+S molecules due to their much higher similarity. In fact, the low QED for LOGRL could also be caused by a correlation between similarity and QED due to the target molecules. This is due to the five trypsin molecules having low QED values (0.277, 0.356, 0.145, 0.159, 0.224) which might result in molecules with higher similarity having lower QED.

We acknowledge that the low performance of Q+S model can be attributed to it being a multi-objective optimization task. By addressing the challenges in multi-objective RL, it would be possible to improve the results. This also compels us to point out the simplicity of the LOGRL model which is able to outperform the Q+S model which is the commonly explored method in prior works (Zhou et al., 2019; Işık & Tan, 2021).

We attain perfect validity and novelty across all our models. This is a consequence of our problem's unique design; the perfect validity is due to the utilization of reaction rules, ensuring that each generated product represents a chemically valid molecule. The perfect novelty stems from having a single target for comparison against the generated set of molecules. None of the generated molecules successfully reaches the target, likely attributed to the absence of the target within the search space defined by our problem, given the initial molecules and set of mined actions. The near-perfect uniqueness is a result of our design, which incorporates an extensive search space. Despite the vastness of possibilities, there exist multiple paths to reaching a molecule, owing to the independent nature of actions when applied to different functional groups of the starting molecule. Consequently, our model excels by seldom generating identical molecules.

We refer the reader to the supplementary material for examples of generated molecules.

## 7  CONCLUSION

We develop a framework for lead optimization as forward synthesis, thereby inducing synthesis-ability as part of the design. We propose LOGRL, an algorithm for this framework using goal-conditioned reinforcement learning that does not use similarity during its training procedure. LOGRL, trained on a different task of source-to-target path prediction is able to generate molecules with significantly higher similarity than RL methods that optimize for similarity. We also suggest a unique generation procedure that allows generating a large number of molecules such that the property optimization task can be performed post-generation as a search.

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
