# Supplementary Material

This document contains supplementary material for the submission titled "A SIMILARITY-AGNOSTIC REINFORCEMENT LEARNING APPROACH FOR LEAD OPTIMIZATION".

## 1. OFFLINE DATASET GENERATION

We generate an offline RL dataset using trajectories with high returns to overcome the sparse reward problem in GCRL. However, given an arbitrary source-target pair, finding if it has high return requires us to know whether there exists a path between the two molecules, which is non-trivial. Instead, if some policy is used to rollout a trajectory, by the virtue of the creation of the trajectory, there exists a path from its starting molecule to the final molecule. The trajectory would therefore have a high return, making it a valid trajectory for the dataset. Following this idea, we select a starting molecule and use a uniform random policy to perform rollouts for $n$ steps to create an $n$-step offline RL dataset. The steps to generate samples for the dataset are as follows:

Start by selecting a source molecule and perform a rollout for $n$ steps using a uniform random policy. This generates a trajectory $\langle m_0, rr_0, m_1, rr_1, ..., m_{n-1}, rr_{n-1}, m_n \rangle$, where $m_t$ represents the molecule at step $t$ and $rr_t$ is the reaction rule applied on $m_t$ to create $m_{t+1}$. Since every $m_t$ has a path to $m_n$ ($t \neq n$), every sub-trajectory $\langle m_t, rr_t, m_{t+1}, rr_{t+1}, ..., m_{n-1}, rr_{n-1}, m_n \rangle$, is also a valid trajectory.

Therefore, for each $t \in [0, n-1]$, we save $\langle s_t, a_t, s_{t+1}, r \rangle$ as the state, action, next state and reward; where the state is a tuple containing the molecule at that state and the target molecule i.e. $s_t = (m_t, m_n)$, action $a_t = rr_t$ and the reward $r = R(s_t, a_t, s_{t+1})$ is calculated as described in the section 4.6 of the main manuscript. This constructs our offline RL dataset.

## 2. TRAINING SETUP

Here we discuss the training details such architecture and hyperparameters for the baseline and LOGRL models.

**Common settings**
We use a Graph Isomorphism Network(GIN) [1] for molecular embeddings and an actor-critic model for learning the policy. The GIN model is pretrained using TorchDrug [2] on Attribute Masking using ZINC dataset [3] with the following parameters: hidden_dims=[128, 128, 128, 128, 128], batch_norm=True, readout="mean", mask_rate=0.15, lr=0.001, batch_size=256, epochs=50. It is allowed to fine-tune during the actor-critic training.

Actor-critic model with the following architecture are used for training: actor with 3 hidden layers each of 256 dimensions with batch_norm and ReLU activation after every layer except the final layer. The critic contains 2 hidden layers each of 256 dimensions with batch_norm and ReLU activation after every layer except the final layer. Adam was used as the optimizer with beta1 and beta2 as 0.9 and 0.999.

**Experiment-specific settings**
For LOGRL experiments, we use the following hyperparameters for training: learning rate(lr) for actor = 0.0003, lr for critic = 0.001, epochs = 50, batch size = 128, and $l = 10$.

For the baselines, we trained using PPO [4] with using the following hyperparameters: lr = 0.0003, total timesteps = $10^7$, batch size = 128, update_epochs=2, clip_coeff=0.2, ent_coeff=0, vf_coeff=0.5.

## 3. ALGORITHMS

The training algorithm (Algorithm 1) is broadly divided into following 3 steps:

1. *Collect positive batch*: In this step, a batch of transitions is sampled from the offline RL dataset.

2. *Collect negative batch*: Here, the states from the positive batch are passed through the actor to obtain predictions. These predictions are used to generate a negative batch using the strategy described in section 4.5 of the main manuscript.

3. *Model update*: The positive and negative batches are passed through the actor and critic models to calculate their respective losses and backpropagate to update their parameters.

The generation algorithm (Algorithm 2) is broadly divided into the following 3 steps:

1. *Filter actions using actor*: Get the closest actions according to Euclidean distance from the actor's prediction.

2. *Filter actions using critic*: Get the actions with highest Q values according to the critic.

3. *Apply*: Apply the filtered actions on the current source to get the next list of sources (or final candidates if last iteration).

## 4. EXAMPLE MOLECULES

Figure S1 shows examples of molecules generated by the LOGRL model for two of the five trypsin inhibitors: Camostat and NAPAMP. The three molecules with highest QED and similarity to target greater than 0.6 are shown, along with additional properties for logP and SA score [**?** ]. The QED values are higher than the target molecule as expected, but all the molecules also ended up having a higher logP than the target molecule, even without us applying a filter for it in our search. Though we do not explore comparison with multi-property optimization works in the scope of this work, the results shown induce confidence in our model to be able to generate lead candidates that satisfy multiple properties. However, we leave this direction of thought for future work. Additionally, we also point out that all the suggested molecules have good SA scores. This is likely due to our design where we ensure the generated molecules follow a system mimicking the actual creation of drugs.

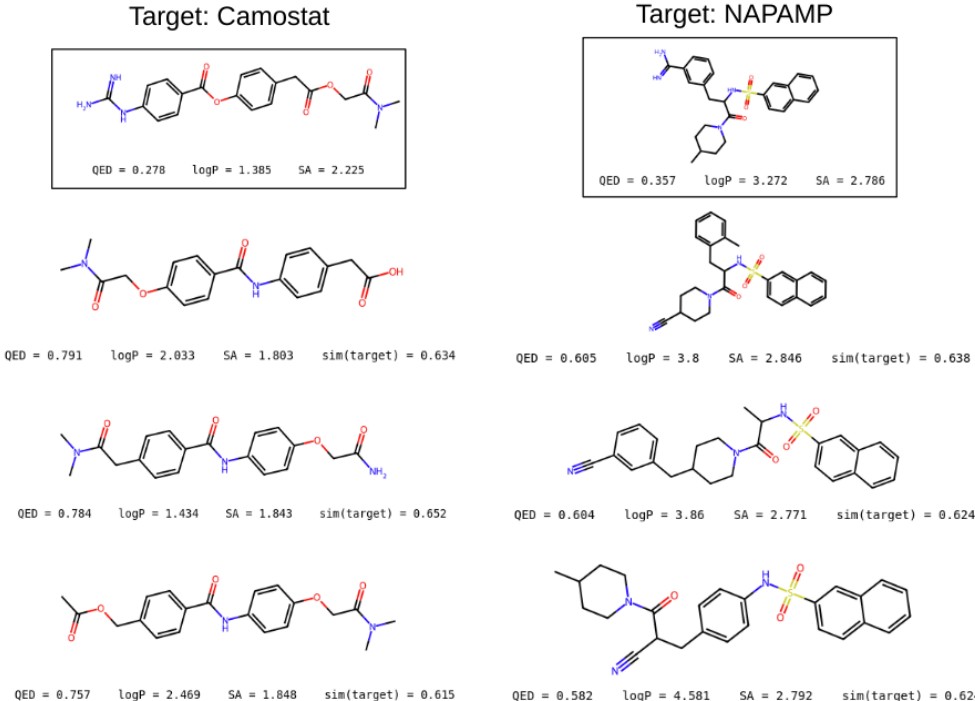

**Fig. S1.** Example of molecules generated by LOGRL for two of the trypsin inhibitors: Camostat and NAPAMP. The three molecules with the highest QED are reported with constraint *sim(target)* > 0.6 along with their logP and SA values.

## 5. MINING REACTION RULES

Here, we describe our mining procedure to extract reaction rules from the USPTO-MIT dataset [5]. First, for each reaction, we created a one-to-one map between each reactant and the product

it transformed into, using the closest molecular weight. Next, we removed duplicates to get unique pairs of reactant-product. Next, we extracted the reaction centres and signatures by subtracting the maximum common substructure from both the reactant and product. Finally, we only preserved those reaction rules where there was a single reaction center as the rules with multiple centers were a very small fraction and often resulted in erroneous cases.