# OpenReview forum: "A SIMILARITY-AGNOSTIC REINFORCEMENT LEARNING APPROACH FOR LEAD OPTIMIZATION"
_ICLR.cc/2024/Conference — Submitted to ICLR 2024_

### Official Review · Reviewer_NNdZ · 2023-10-27

**Soundness:** 1 poor
**Presentation:** 3 good
**Contribution:** 2 fair
**Rating:** 3
**Confidence:** 3

**Summary:**

The paper tries to tackle a challenging problem in drug discovery, where it is common to optimize a lead compound to remove deficiencies and maintaining the favorable properties.
They highlight the challenges of using reinforcement learning based on similarity metrics to define certain constrains on the optimized compound, which potentially can introduce a bias in the generative process.
Therefore, the authors propose a so call similarity agonistic reinforcement learning approach and remove the dependency on the similarity metric as additional constrain for optimization. This is achieved by goal-conditioned reinforcement learning.

**Strengths:**

In my opinion the paper has the following strengths:

-	To the best of my knowledge the idea of using complete molecules as goal (for goal conditioned reinforcement learning), as the authors propose it, is novel.
-	In general, the method section is well explained with minor exceptions.
-	Using reaction rules partially circumvents a general problem in generative models for drug discovery, namely a significant part of generated molecules are difficult to synthesize in the lab hindering a fast-pace early stage drug discovery program. The use of reaction rules conditions the generative model to generate more chemically plausible molecules with a direct synthesize path.
-	The method seems to improve upon their baseline on all experiments.

**Weaknesses:**

Lead optimization in drug discovery is an important and difficult task. I have difficulty accepting the method as an invention or improvement for lead optimization. For my understanding lead optimization is a much more complicated process than purely looking on QED score or a similarity score, which the authors didn’t investigated.

In general, the paper would gain strength if the authors would compare their method against more recent methods in generative design and more properties other than QED. Especially, the baseline seems to be quite weak with all the efforts recently put into improving generative methods.
For example, the author could have a look at a standardized benchmark, e.g. [1].
This would strength their method and would help to better showcase the potential
improvement compared to other methods.
The authors might also consider comparing their methods against other methods in the domain of scaffold hopping, e.g. [3].

The second contribution of their paper as stated on page 2, says:
“we propose a search strategy…”
Could the authors elaborate more on the search strategy? In case it is just generating thousands of molecules and sorting them based on a score, this seems to me not like a novel strategy.

I very much like the idea of using reaction rules, although not completely novel, e.g. [2]. I think a more detailed description how exactly they mine the reaction rules and a better description of the reaction dataset in general would help the reader to better understand the topic. It doesn’t have to be in the main text.

I had trouble understanding the last paragraph of section 4.6., “we found that under the condition G(a_t leads to g)…”. Could the authors elaborate a little bit more on the issue they observed?

To summarize, although certain ideas are interesting and in some sense novel, I am hesitant to accept the paper mainly because of in my opinion a weak experiment section. The paper doesn’t showcase a technique for lead optimization, which is much more complicated than what is investigated in the paper. Also, claims like:
“Though we do not explore comparison with multi-property optimization works in the scope of this work, the results shown induce confidence in our model to be able to generate lead candidates that satisfy multiple properties.” Sec. 6,
seem to be too strong for the experiments considered.


[1] Wenhao Gao, Tianfan Fu, Jimeng Sun, and Connor W. Coley. Sample Efficiency Matters: A
Benchmark for Practical Molecular Optimization, October 2022.

[2] Tianfan Fu, Wenhao Gao, Connor W. Coley, Jimeng Sun, Reinforced Genetic Algorithm for Structure-based Drug Design, 2022.

[3] Krzysztof Maziarz et al. LEARNING TO EXTEND MOLECULAR SCAFFOLDS WITH STRUCTURAL MOTIFS, 2022.

**Questions:**

-	Did I understand it correctly that the offline dataset just contains molecules randomly put together using the reaction rules, so potentially not chemically plausible at all?
-	My understanding of actor-critic reinforcement learning is to use the output of the critic for the loss of the actor. From eq. (1) and (2) this seems not the be the case, could the authors elaborate a little bit?

---

> ### Author Response · Authors · 2023-11-19
> **Response to Reviewer NNdZ [1/2]**
>
> Thank you for the review. To address common concerns by the reviewers, we have added a Common Response (https://openreview.net/forum?id=rjLgCkJH79&noteId=AeabWUzjwD). We respond to the reviewer's comments below along with reference to the Common Response where needed.
>
>
> > “Lead optimization in drug discovery is an important and difficult task. I have difficulty accepting the method as an invention or improvement for lead optimization. For my understanding lead optimization is a much more complicated process than purely looking on QED score or a similarity score, which the authors didn’t investigated.”
>
> We refer the reviewer to Common Response (1).
>
> As mentioned, our method is a generalized approach for generating alternate candidates to the lead molecule. Therefore, we have not explored the subsequent step of docking or binding affinity in the context of our work.
>
>
>
> > “In general, the paper would gain strength if the authors would compare their method against more recent methods in generative design and more properties other than QED.“
>
> We refer the reviewer to Common Responses (2) and (3). Results for the added baseline and metrics can be found in the Results Table in the Common Response.
>
>
>
> > “The second contribution of their paper as stated on page 2, says: “we propose a search strategy…” Could the authors elaborate more on the search strategy? In case it is just generating thousands of molecules and sorting them based on a score, this seems to me not like a novel strategy.”
>
> The reviewer is referring to the following statement in the paper:
>
> “We propose a search strategy that separates the property optimization from the training and offloads it as a post-curation process, thereby simplifying the task of learning.”
>
>
> We acknowledge the comment made by the reviewer. The novelty comes from the idea of using the search strategy to offload the property optimization from the training procedure due to the problems associated with multi-objective optimization.
>
> To explain further, we take the example of the reward function used in MolDQN [1] for multi-objective optimization:
>
> $r_t = w \times SIM(m_t, m_0) + (1-\omega) \times QED(m_t)$
>
> The optimal value of ‘w’ in this multi-objective case is hard to determine quantitatively during hyperparameter tuning. In fact, the notion of optimality may be task-dependent and may very likely not be well understood for most tasks. If the user is not satisfied with the generated molecules, a new model would need to be trained with a different ‘w’. As opposed to this, in our work, the model would not need to be re-trained. Only the much cheaper search strategy needs to be employed to generate a large number of molecules that can be filtered to retain desired properties. Other multi-objective optimization challenges, such as Pareto dominance, conflicting objectives, etc are also avoided in our method.
>
>
>
> >“I very much like the idea of using reaction rules, although not completely novel, e.g. [2]. I think a more detailed description how exactly they mine the reaction rules and a better description of the reaction dataset in general would help the reader to better understand the topic. It doesn’t have to be in the main text.”
>
> We thank the reviewer for the acknowledgment and suggestion. We have added the details to the supplementary material.
>
>
>
> >“I had trouble understanding the last paragraph of section 4.6., “we found that under the condition G(a_t leads to g)…”. Could the authors elaborate a little bit more on the issue they observed?”
>
> Thank you for the question. We had noticed that based on different values of the negative rewards that we chose, the policy would either converge or diverge.
>
>
>
> To recap,
>
> -   In our method, we select a fixed number L of negative return trajectories per high return trajectory.
>
> -   The policy gradient loss moves the function away from low-return areas and closer to high-return areas.
>
>
> Therefore, if the magnitude of the negative reward were high, the cumulative magnitude of gradient from the negative samples in a batch would overwhelm the magnitude of positive samples, causing it to diverge. The condition we mentioned works as a normalizing factor for the negative component of the loss and was essential for reward design in our case to ensure convergence.

---

> ### Author Response · Authors · 2023-11-19
> **Response to Reviewer NNdZ [2/2]**
>
> **Questions:**
>
> >Q1. Did I understand it correctly that the offline dataset just contains molecules randomly put together using the reaction rules, so potentially not chemically plausible at all?
>
> We are unsure if we understood the question correctly. The trajectories in the offline dataset are indeed put together using a random policy **given the reaction rules**. This still makes them chemically feasible though.
>
> We elaborate for further clarification: Say we are given a start molecule. To generate a trajectory for the offline dataset, first, a reaction rule will be selected randomly among those that are applicable to it. This will result in a product which is chemically feasible. even though it is the result of a random sampling. By repeating this step multiple times, we generate a trajectory for our dataset using a random policy where each molecule is chemically feasible.
>
>
>
> >Q2: My understanding of actor-critic reinforcement learning is to use the output of the critic for the loss of the actor. From eq. (1) and (2) this seems not the be the case, could the authors elaborate a little bit?
>
> The reviewer has noticed correctly. Due to our unique setup, we were able to use calculated returns to update the actor, thus not requiring the higher variance estimation by the critic.
>
>
>
> The critic is still a crucial part of our method. Firstly, it provides additional gradients to update the embedding module, resulting in higher-quality representations. Secondly, it is used in the generation procedure to sort the actions suggested by the policy.

---

> > ### Comment · Reviewer_NNdZ · 2023-11-22
> >
> > I very much appreciate the effort the authors put into the rebuttal and thank you for answering my questions.
> >
> > Although, I think certain parts of the approach are very interesting, for example, the large action space, the experimental section is still not convincing to me. This is among other things connected to the metrics considered. Almost all methods considered have a validity, uniqueness, and novelty score of 1. The QED and similarity score alone are not convincing.
> >
> > I strongly recommend the authors to consider other benchmark systems. As a suggestion maybe they can apply their method to scaffold hopping or test on standard benchmark systems like a Dopamine Type 2 Receptor Case study (c.f. [1]).
> > Therefore, I will leave my score as is.
> >
> > [1] "Augmented Memory: Capitalizing on Experience Replay to Accelerate De Novo Molecular Design", Jeff Guo and Philippe Schwaller, (2023).

---

### Official Review · Reviewer_PmbC · 2023-10-31

**Soundness:** 1 poor
**Presentation:** 2 fair
**Contribution:** 1 poor
**Rating:** 3
**Confidence:** 3

**Summary:**

This work presents LOGRL, a unique approach to lead optimization using a goal-conditioned reinforcement learning framework. Given an expert dataset, this work trains a goal-conditioned policy with binary reward shaping, treating reaction rules as actions. Then, LOGRL compares Tanimoto similarity and QED of generated molecules with two baselines using an online RL method, which is PPO.

**Strengths:**

- The paper is well-written and presents clearly.
- The paper demonstrates comprehensive related work.

**Weaknesses:**

The experimental comparison in this paper raises some concerns regarding fairness and appropriateness. The authors compare their proposed off-line Reinforcement Learning (RL) policy with on-line RL baselines. This comparison between on-line and off-line RL algorithms seems somewhat unconventional. Moreover, it's unclear whether the on-line RL baselines, such as the S model and Q+S model, employ an expert dataset similar to LOGRL. If they do not utilize expert data, this could introduce an unfair advantage to LOGRL, as it relies on additional expert data. It would be beneficial to see how LOGRL performs when compared to baselines that also use the same expert dataset.

Additionally, I suggest exploring the possibility of supervised learning in this context. The authors assume access to a substantial amount of expert dataset containing high-reward samples. In such a scenario, imitation learning often outperforms offline RL. It would be valuable to understand why the authors chose offline RL over supervised learning, given the abundance of expert data.

The paper employs policy gradient, which typically assumes that the training policy and the behavior policy are aligned, making it an on-policy approach. The suitability of using a policy gradient in an offline RL setup is a point of concern. It would be helpful to see more discussion and justification regarding the use of an on-policy algorithm like policy gradient in this context.

Finally, it would be interesting to know if the proposed method is capable of generating diverse outputs. One potential concern is whether the method might collapse and generate a single output, as there doesn't appear to be a regularizer that can control all possible outputs directed toward the target molecule. Exploring the diversity of outputs and addressing this potential issue would strengthen the paper.
Overall, while the paper presents a promising approach, addressing these concerns and providing more clarity would enhance the quality of the work and its relevance in the field of machine learning and RL.

---

minor

Typo in Section 4.5 line3: in the training batch, batchwe

**Questions:**

See Weakness section

---

> ### Author Response · Authors · 2023-11-19
> **Response to Reviewer PmbC**
>
> Thank you for the review. To address common concerns by the reviewers, we have added a Common Response (https://openreview.net/forum?id=rjLgCkJH79&noteId=AeabWUzjwD). We respond to the reviewer's comments below along with reference to the Common Response where needed.
>
> > “The authors compare their proposed off-line Reinforcement Learning (RL) policy with on-line RL baselines. This comparison between on-line and off-line RL algorithms seems somewhat unconventional.”
>
> We agree with the reviewer that the comparison of offline RL with online RL baseline might be unconventional. However, we would like to clarify that our intended comparison is not between online and offline RL methods but instead between our binary reward method and prior distance-based reward methods (that use molecular similarity as rewards). Due to binary rewards leading to severe sparsity in our problem setup, we have adopted the use of binary rewards in an offline setting.
>
> We have added another baseline for Random Search and refer the reviewer to the Results Table in our Common Response.
>
> >“Moreover, it's unclear whether the on-line RL baselines, such as the S model and Q+S model, employ an expert dataset similar to LOGRL. If they do not utilize expert data, this could introduce an unfair advantage to LOGRL, as it relies on additional expert data.”
>
> The baselines are completely online and, hence do not employ any expert data. Since the offline dataset was not collected using an expert policy or a human, we disagree with the reviewer to call it “expert data”. The data available to the offline algorithm is a result of running a random policy in the same underlying environment as the online RL baseline and relabelling the rewards. The procedure is very trivial and should not provide any extra information to the offline RL algorithm.
>
> >“Additionally, I suggest exploring the possibility of supervised learning in this context. The authors assume access to a substantial amount of expert dataset containing high-reward samples. In such a scenario, imitation learning often outperforms offline RL.”
>
> We thank the reviewer for the suggestion. We had explored behavior cloning(BC) during our early experiments and found offline RL to significantly outperform BC. Results from those experiments are given below. We evaluated BC vs our model on trajectories of different lengths (steps) - 1, 2 and 5 and calculated the top-k accuracy of the model. Top-k accuracy here indicates the percentage of test samples for which the model ranked the positive action within the top k of its predictions. The top k predictions are the actions suggested in a single step of the generation procedure described in Algorithm 2 in the manuscript. The results for those experiments are given below.
>
>
> | Steps | Model | top-10 | top-5 | top-1 |
> |:-----:|:-----:|:------:|:-----:|:-----:|
> |   1   |   BC  |  96.05 | 89.48 | 73.13 |
> |       | LOGRL |   100  |  100  |  100  |
> |       |       |        |       |       |
> |   2   |   BC  |  35.51 | 27.45 |   13  |
> |       | LOGRL |  96.1  | 95.62 | 95.22 |
> |       |       |        |       |       |
> |   5   |   BC  |  32.42 | 23.92 |  9.37 |
> |       | LOGRL |  84.13 | 84.07 | 84.02 |
>
> > “Finally, it would be interesting to know if the proposed method is capable of generating diverse outputs. One potential concern is whether the method might collapse and generate a single output, as there doesn't appear to be a regularizer that can control all possible outputs directed toward the target molecule. Exploring the diversity of outputs and addressing this potential issue would strengthen the paper.“
>
> We thank the reviewer for pointing it out. We have added an evaluation of the uniqueness of generation along with metrics for validity and novelty. Uniqueness is a measure of how unique are the molecules generated; 1 indicates all molecules are unique (or perfect diversity) and 0 indices all the generated molecules are the same (no diversity). All models, including our baselines, achieve near-perfect uniqueness scores. However, a major contribution to this metric would be a result of our design of the problem; due to the incredibly large search space and the generation procedure promoting uniqueness. That said, there do exist multiple paths for generating the same molecule, therefore the models are also effective in not reproducing the same paths repeatedly. We refer the reviewer to the Results Table in the Common Response for the metric scores.
>
>
>
> > “Typo in Section 4.5 line3: in the training batch, batchwe”
>
> Thank you for pointing it out. We have corrected it.

---

### Official Review · Reviewer_TrZv · 2023-11-01

**Soundness:** 2 fair
**Presentation:** 3 good
**Contribution:** 2 fair
**Rating:** 5
**Confidence:** 4

**Summary:**

In this study, a new lead optimization method, LOGRL, is proposed. This method uses offline reinforcement learning to train the model how to optimize molecular structures to get closer to the target structures (goal-conditioned reinforcement learning). Furthermore, a set of reactions is used to ensure the synthetic accessibility of the generated structures. The beam search algorithm is used, which helps in obtaining a diverse set of modified structures that meet desired properties. LOGRL is compared against two RL baselines and achieves promising results in optimizing molecules towards the target structures, both in terms of similarity and drug-likeness defined by QED.

**Strengths:**

- The method is presented in a very clear way. The background section provides all the basics that are required to understand the method.
- Offline reinforcement learning is used to avoid sparse rewards when navigating the vast chemical space.
- The goal-conditioned reinforcement learning is used to guide the generative process, which in my opinion is the main novelty of the paper. This way, similarity measures are no longer needed to train the model.
- Reaction rules extracted from the USPTO-MIT dataset are used to ensure the synthesizability of the generated molecules, which is important for proposing high-quality results.

**Weaknesses:**

- The significance of the work is not clear. The method is trained to optimize molecules towards the target structures, but I am unsure if I understand how this model could be used in practice. Usually, the goal of lead optimization is to improve a set of molecular properties without impacting binding affinity. In the presented setup, the optimization changes the structure of lead candidates to make them more similar to known drugs, which oftentimes is undesired because only novel structures can remain outside the patented chemical space.
- The experimental section seems very preliminary. Only two RL baselines were trained, and there is no comparison with other state-of-the-art methods in molecular optimization. The evaluation metrics used in the experiments are very simple and do not show if the proposed method can optimize any molecular properties or at least retain high binding affinity. The Authors claim that their search strategy separates property optimization from training, but the results of the optimization are not presented. Additionally, all methods were run only once (if I understand correctly), and the results can be hugely impacted by random initialization, especially for online RL methods like the baselines. I would strongly suggest running these methods multiple times and providing confidence intervals for the evaluation metrics.
- (minor) I think the Authors could consider comparing their approach to the simpler, yet conceptually similar, Molpher model [1]. In Molpher, a trajectory between two molecules is found by an extensive search (not RL-based) of possible reaction-based structure mutations. The motivation of that paper is also different, Molpher was proposed for effective chemical space exploration.

[1] Hoksza, David, et al. "Molpher: a software framework for systematic chemical space exploration." Journal of cheminformatics 6.1 (2014): 1-13.

**Questions:**

1. What is the success rate of molecular optimization using LOGRL? Can you find many well-optimized molecules in the post-training filtering step, or do you think additional RL objectives could improve these properties significantly?
2. What are the real-life applications of this optimization algorithm? Can it be used for other optimization problems besides lead optimization (see the problems I mentioned in the "Weaknesses" section)?
3. In Section 3.2, two GCRL methods are described. Did you try the other method and if so, could you provide the comparison results?

---

> ### Author Response · Authors · 2023-11-19
> **Response to Reviewer TrZv**
>
> Thank you for the review. To address common concerns by the reviewers, we have added a Common Response (https://openreview.net/forum?id=rjLgCkJH79&noteId=AeabWUzjwD). We respond to the reviewer's comments below along with reference to the Common Response where needed.
>
> > "The significance of the work is not clear.  The method is trained to optimize molecules towards the target structures, but I am unsure if I understand how this model could be used in practice. Usually, the goal of lead optimization is to improve a set of molecular properties without impacting binding affinity."
>
> We refer the reviewer to Common Response (1).
> Having not considered a target receptor, we did not discuss binding affinity. We thank the reviewer for pointing it out. We acknowledge that it would be an important step after our method for further curation.
>
> > "Only two RL baselines were trained, and there is no comparison with other state-of-the-art methods in molecular optimization."
>
> We refer the reviewer to Common Response (2).
>
> > "The evaluation metrics used in the experiments are very simple."
>
> We refer the reviewer to Common Response (3).
>
> A table with additional baseline and metrics is present in the "Results Table" section of Common Response.
>
> **Questions:**
>
> > Q1 (A): What is the success rate of molecular optimization using LOGRL?
>
> Without docking, the notion of success rate depends on the user. For instance, for our chosen examples of trypsin inhibitors, among 10000 molecules generated by LOGRL, 73 had QED > 0.7.
>
> > Q1 (B): Can you find many well-optimized molecules in the post-training filtering step?
>
> It is possible to find many optimized molecules in the post-training filtering step. Towards this effect, we can simply increase the breadth or depth of the search, which results in a larger number of molecules.
>
> > Q1 (C): Do you think additional RL objectives could improve these properties significantly?
>
> Additional RL objectives would surely improve the properties of generated molecules. Unfortunately, this has several drawbacks, some are mentioned in our paper. Firstly, for each combination of properties, there would need to be a separate model trained. Secondly, problems related to multi-objective optimization: additional cost of training due to extra hyperparameters, Pareto dominance, conflicting objectives, etc.
>
> Our method avoids these problems by separating molecular search and property search into two different steps.
>
> > Q2: What are the real-life applications of this optimization algorithm? Can it be used for other optimization problems besides lead optimization (see the problems I mentioned in the "Weaknesses" section)?
>
> The proposed algorithm can be used in offline RL settings with a large discrete action space, such as recommender systems and language generation.
>
> > Q3: In Section 3.2, two GCRL methods are described. Did you try the other method and if so, could you provide the comparison results?
>
> The two methods described are binary rewards and distance-based rewards. Our proposed method uses binary rewards, while the online RL baselines use distance-based rewards. To our understanding, the comparison requested is present in the paper.
>
> We would also like to clarify that similarity-based rewards are a type of distance-based reward.

---

### Author Response · Authors · 2023-11-19
**Common Response to Reviewers [1/2]**

We thank the reviewers for their comments. Their insightful comments have been invaluable in enhancing the quality and clarity of our work. We have carefully addressed each of the concerns raised in the reviews, and we believe that the revisions made have significantly strengthened the paper. We address some of the common concerns of the reviewers here.


 ***(1) Lead optimization is a much more complicated process than what our method addresses.***

Our method does not solve lead optimization entirely. We have presented it as a generalized approach towards proposing alternatives to lead compounds which is a crucial part of lead optimisation.

We have updated the introduction section in the paper with the above clarification.

***(2) Lack of baselines.***

Our two RL baselines are the most “fair” comparisons, in our opinion. A direct comparison with most other methods would be unfair since their proposed methods were conceptualized for *de novo drug design*, and only their application was demonstrated on lead optimization (often as a constrained optimization task with similarity constraints). **Our work, to our knowledge, is the first generalized approach to lead optimization that does not apply to *de novo drug design*.** This leads to a couple of differences that would be hard to bridge:

1.  Having a target molecule - In *de novo drug design*, there is no notion of a target molecule. Most prior methods modify the lead candidate itself to find novel structures and apply a similarity constraint to limit the extent of modifications. These approaches would not work in our setting where we modify a small molecule towards the lead candidate.




As mentioned in the above point, since most prior methods modify the lead compound to discover novel compounds, the synthesizability of the molecules is often questionable. Towards that effect, we employ a retrosynthesis-like approach of using “reaction rules” that make our generation process in line with natural chemical reactions, resulting in our much more feasible molecules. To constitute our action space with these rules, it leads us to the second difference:



2.  Large discrete action space - Our action space is about 85K, whereas most papers consider action spaces in orders of tens or hundreds at most. This results in most prior RL methods proposed for molecular optimization being unsuitable for our task.




Considering the above points, the closest work to ours that we found was [1], which also closely resembles the baselines used in our work. Though, since [1] was also proposed for *de novo drug design*, we note the major differences here - change of state space to incorporate 2 molecules (source and target) as opposed to their single molecule, and the use of a single policy network as opposed to their two-fold setup.



**We realize the lack of baselines would still be a weakness; therefore, inspired by [1] and Molpher [2] (suggested by the reviewer TrZv), we have added another baseline that uses Random Search. The table with the additional baseline is given below and has also been added to the main paper.**


The procedure of Random Search is identical to [1]: initially, a random molecule is chosen $m_0$ and a random action is applied on it to produce $m_1$ . This is repeated until a termination condition of the trajectory is met; either the maximum length of trajectory is achieved or no actions are applicable on the molecule. This is repeated 200 times for each trypsin inhibitor target in our work.

***(3) Lack of metrics.***

As described above, since our task is fairly different from *de novo drug design*, most conventional benchmarks such as Guacamol [3] are not well-suited for it. However, since reviewer PmbC specifically raised concerns about diversity(or uniqueness), **we have added the following distribution-learning metrics from Guacamol [3]: Validity, Uniqueness, and Novelty.** The other distribution-learning metrics: KL divergence and FCD are inapplicable since each molecule is generated against a single target. Due to the lack of a target distribution, similarity may be used as a surrogate measure for KL divergence and FCD here.

**Results Table:**

Below is the table with additional baseline and metrics. The table has also been updated in the main paper along with further analysis of the additional baseline and metrics. We present the table below for the convenience of the reviewers.

---

> ### Author Response · Authors · 2023-11-19
> **Common Response to Reviewers [2/2]**
>
> | Method        | Molecules | Mean sim  | QED Min   | QED Mean  | QED Max   | Validity | Uniqueness | Novelty |
> |---------------|-----------|-----------|-----------|-----------|-----------|----------|------------|---------|
> |               | 1         |     0.477 |     0.483 |     0.483 |     0.483 |        1 |          1 |       1 |
> |               | 10        |      0.45 |     0.353 |     0.448 |     0.559 |        1 |          1 |       1 |
> | Random Search | $10^2$      |     0.417 |     0.109 |     0.418 |     0.841 |        1 |          1 |       1 |
> |               | $10^3$      |     0.377 |     0.041 |     0.385 |     0.841 |        1 |          1 |       1 |
> |               | $10^4$      |     0.333 |     0.022 |     0.316 |     0.929 |        1 |          1 |       1 |
> |               |           |           |           |           |           |          |            |         |
> |               | 1         |     0.694 |     0.289 |     0.289 |     0.289 |    **1** |      **1** |   **1** |
> |               | 10        |     0.618 |      0.12 |     0.201 |     0.289 |    **1** |      **1** |   **1** |
> | S model       | $10^2$      |     0.554 |     0.049 |     0.206 | **0.712** |    **1** |       0.99 |   **1** |
> |               | $10^3$      |     0.499 |     0.024 |     0.262 | **0.923** |    **1** |      0.999 |   **1** |
> |               | $10^4$      |     0.439 |     0.013 | **0.302** | **0.946** |    **1** |     0.9988 |   **1** |
> |               |           |           |           |           |           |          |            |         |
> |               | 1         |     0.703 | **0.294** | **0.294** | **0.294** |    **1** |      **1** |   **1** |
> |               | 10        |     0.652 | **0.264** | **0.401** | **0.681** |    **1** |      **1** |   **1** |
> | Q+S model     | $10^2$      |      0.59 | **0.102** | **0.299** |     0.681 |    **1** |      **1** |   **1** |
> |               | $10^3$      |     0.523 |     0.029 |     0.226 |     0.845 |    **1** |      0.999 |   **1** |
> |               | $10^4$      |      0.45 |     0.013 |     0.261 |     0.942 |    **1** |     0.9997 |   **1** |
> |               |           |           |           |           |           |          |            |         |
> |               | 1         | **0.791** |     0.262 |     0.262 |     0.262 |    **1** |      **1** |   **1** |
> |               | 10        | **0.761** |     0.127 |     0.244 |     0.321 |    **1** |      **1** |   **1** |
> | LOGRL         | $10^2$      | **0.704** |     0.088 |     0.294 |     0.624 |    **1** |      **1** |   **1** |
> |               | $10^3$      | **0.634** | **0.032** |   **0.3** |     0.791 |    **1** |      **1** |   **1** |
> |               | $10^4$      |  **0.56** | **0.018** |     0.291 |     0.889 |    **1** | **0.9999** |   **1** |
>
> #### Table contains three baselines: Random Search, S model (similarity rewards), Q+S model (QED+sim rewards) and our model LOGRL. Evaluation is performed for the top $10^n$ molecules based on highest similarity ("Molecules" column). The rest of the columns are the metrics used for evaluation and are self-explanatory.
>
>
> [1] Gottipati, Sai Krishna, et al. "Learning to navigate the synthetically accessible chemical space using reinforcement learning." International conference on machine learning. PMLR, 2020.
> [2] Hoksza, David, et al. "Molpher: a software framework for systematic chemical space exploration." Journal of cheminformatics 6.1 (2014): 1-13.
> [3] Brown, Nathan, et al. "GuacaMol: benchmarking models for de novo molecular design." Journal of chemical information and modeling 59.3 (2019): 1096-1108.

---

### Meta-Review · Area_Chair_hJig · 2023-12-14

**Metareview:**

This paper introduces an RL algorithm which allows for generation of molecules in the neighborhood of an existing lead, without the need to explicitly specify a molecular similarity function. This is a nice idea, and the paper presents it quite clearly. However, the reviewers had a number of concerns regarding the appropriateness of baselines (online/offline RL settings), the role of RL relative to supervised/imitation learning, and the general suitability of the approach for difficult lead generation/refinement tasks. As such at the moment the consensus among reviewers is this is not yet ready for publication at ICLR.

**Justification For Why Not Higher Score:**

All reviewers suggested to reject, based primarily on concerns regarding evaluation.

**Justification For Why Not Lower Score:**

N/A

---

### Decision · Program_Chairs · 2024-01-16

Reject